# Potato Microbiome: Relationship with Environmental Factors and Approaches for Microbiome Modulation

**DOI:** 10.3390/ijms25020750

**Published:** 2024-01-06

**Authors:** Ivan S. Petrushin, Nadezhda V. Filinova, Daria I. Gutnik

**Affiliations:** Siberian Institute of Plant Physiology and Biochemistry, Siberian Branch of the Russian Academy of Sciences, Irkutsk 664033, Russia; filinova_nv@mail.ru (N.V.F.); daria_gutnik@mail.ru (D.I.G.)

**Keywords:** *Solanum tuberosum*, microbiome, rhizosphere, common scab, potato, pests

## Abstract

Every land plant exists in a close relationship with microbial communities of several niches: rhizosphere, endosphere, phyllosphere, etc. The growth and yield of potato—a critical food crop worldwide—highly depend on the diversity and structure of the bacterial and fungal communities with which the potato plant coexists. The potato plant has a specific part, tubers, and the soil near the tubers as a sub-compartment is usually called the “geocaulosphere”, which is associated with the storage process and tare soil microbiome. Specific microbes can help the plant to adapt to particular environmental conditions and resist pathogens. There are a number of approaches to modulate the microbiome that provide organisms with desired features during inoculation. The mechanisms of plant–bacterial communication remain understudied, and for further engineering of microbiomes with particular features, the knowledge on the potato microbiome should be summarized. The most recent approaches to microbiome engineering include the construction of a synthetic microbial community or management of the plant microbiome using genome engineering. In this review, the various factors that determine the microbiome of potato and approaches that allow us to mitigate the negative impact of drought and pathogens are surveyed.

## 1. Introduction

Potato (*Solanum tuberosum*), an essential food crop worldwide after rice, maize and wheat [1], is highly susceptible to abiotic stresses (salinity and drought), pests and pathogens. Among the negative biotic factors, bacterial [2] and fungal [3] diseases and other pests reduce the yield and storage capacity more than others. Particular attention is paid to the following pests: nematodes [4], the Colorado potato beetle [5], the potato tuber moth [6], aphids [7] and leaf miners [8]. All these factors certainly influence the diversity and composition of the microbiome. The potato microbiome, as in other plants, performs important functions to support their growth and development. The initial microbiome originates from seeds or tubers with tare soil. After planting, potato seed tubers are colonized by soil microbes that can impact the development of the plant [9]. Secondary metabolites allow plants to recruit specific types of microbes to reduce the negative effects of drought, salinity and pathogens, including disease-induced changes in potato plants [10]. Some researchers call this phenomenon “cry for help” [11,12]. At the same time, the composition and diversity depend on the potato plant compartment: the most diverse microorganisms are located in the soil and rhizosphere, while the microbiome of the plant upper parts is less diverse [13]. Such a difference in diversity between compartments is due to the selective forces of the plant host, which are much stronger in the endosphere than in the rhizosphere.

Many metagenomics studies show the difference in composition of the microbial community in the plant niches (leaves, roots, seeds and rhizosphere) [14]. This has been confirmed for potato plants, where bacterial α-diversity and composition of endophytes in roots, stems and tubers differ significantly [15]. Potato plants have a specific part, tubers, and the soil near tubers as a sub-compartment is usually called the “geocaulosphere” [16]. At first sight, the microbial community of the geocaulosphere should differ from that of the rhizosphere because nutrients are excreted by plant roots, but not by tubers. However, some authors have reported an alternative scenario: the microbiome composition of the rhizosphere and geocaulosphere looks similar (when comparing their bacterial phyla composition) [17,18]. It is important to focus on the surface of roots, known as the rhizoplane, and distinguish it from the rhizosphere. A recent report suggests that the species diversity of fungal communities in the potato rhizoplane is considerably lower than that in the rhizosphere [19].

In the overall microbial community of a plant, the core microbiome and the microbiome consisting of satellite species can be distinguished. The core microbiome of plants includes key taxa of microorganisms that are necessary for normal functioning of the host organism. It is formed in the process of evolutionary mechanisms by the selection of microbial taxa containing the necessary genes [20]. The main representatives of the potato core microbiome are *Bradyrhizobium*, *Sphingobium* and *Microvirga*, and the most abundant genera in the rhizosphere are *Lentzea* and *Streptomyces* [10,21], which is confirmed by a very recent large-scale microbiome study across the US regions [22]. Additionally, Custer et al. reported that in the rhizosphere, members of the phyla Actinomycetota and Pseudomonadota showed the highest relative abundances, with *Lentzea* and *Streptomyces* being the most abundant genera [23]. Part of the core microbiome is represented by so-called “hub microorganisms” that influence community structure through strong biotic interactions with the host and other microbial species [24].

These are essential species that can directly and indirectly influence the microbiome composition and act as mediators between the plant and the associated microbiome [25]. These microorganisms have a regulatory effect on the microbial interaction network and their absence leads to microbiome disruption [26]. To modify its adaptive potential, the host plant influences the structure of the associated microbiota via the core microorganisms by regulating intermicrobial interactions. The abundance of “hub microorganisms” is controlled by regions of the host genome involved in carbohydrate metabolism and stress responses [27,28]. Taxa that occur in small amount are called satellite taxa [29], which play an important role in plant adaptation to different habitat conditions. Their composition varies depending on the habitat. Plant microbiomes form a complex interconnected microbial network [30]. Ecological network analysis allows us to define the major microbial species that may have the greatest impact on diversity of the microbial community and its functions [31], e.g., bacterial pathogen resistance of potato plants (*Ralstonia solanacearum*) [32].

The plant microbiome is a dynamic, rather unstable structure, which is influenced by many factors including both biotic and abiotic. Various host, microbial and environmental factors have an effect on the community composition and diversity of the plant microbiome. Any external stress such as a lack of moisture, overwatering and application of chemicals changes the physico-chemical properties of the soil. This consequently affects the microbial association in the soil. Temperature, UV radiation, bacterial and fungal pathogens and insect pests also influence the microbiome composition. In the rhizosphere microbiomes, the same types of microorganisms such as Acidobacteriota, Pseudomonadota, Planctomycetota, Actinomycetota, Bacteroidota and Bacillota are most often dominant among prokaryotes [33]. During plant growth and under the influence of certain factors, the qualitative composition of the potato plant microbiome may remain intact, while the quantitative ratio of microorganisms may be changed [21]. A large body of research is currently focused on the impact of environmental factors on the structure and dynamics of the plant microbiome [25,34,35]. However, the basic mechanisms underlying the formation of the microbiome and their effects on the host plant are still poorly understood. Existing reviews related to the potato plant microbiome tend to focus on specific aspects of potato crop production: soil amendment [36], tillage practices [37], sustainable agrosystems [38], straw mulching practices [39] and disease control using micronutrients and high solubilization capabilities [40]. Most of the reviews we examined did not adequately address the interaction between the potato plants and microbes, nor did they explore the diversity and structural changes in the potato microbiome under the influence of biotic and abiotic factors. Despite the pool of studies on inoculation of potato plants with various microbial mixtures, some of which are discussed in Section 7, there are no reviews dedicated to this topic. At the same time, there are reviews devoted to the microbiome of less significant plant species and families such as citrus [41], apple [42] and drought-resistant legumes [43].

During the whole agricultural history, many approaches have been used to obtain plants with desirable traits. As the microbiome is an essential component of plant health, an increasing focus is on the use of biologicals to modulate the microbiome in crop production. To overcome the negative impact of various factors, the microbial communities of potato plants could be changed (modulated) by inoculation (treatment of soil or tubers with a microbial mixture) [44]. The use of plant growth promoting rhizobacteria (PGPR) in agriculture is becoming the common practice, while rhizobacteria promote the ability to use alternative nutrient uptake [45], mitigating the environmental stress and suppressing pathogen invasion. While PGPR have a wide metabolic potential, the role of the fungal community cannot be neglected. The arbuscular mycorrhizal fungi (AMF) could enhance growth in many plant species, including potato, and interfere with plant–pathogen interactions by the induction of beneficial microbe-associated molecular patterns [46]. The application of PGPR and AMF for potato plants is discussed further in Section 7.

Some approaches to obtain plant microbiomes with particular features are called “microbiome engineering” [47], which involves deep changes in the microbiome. The first studies using such an approach were performed on the model plant *Arabidopsis*, where the plant selected bacteria that help to change the leaf biomass and flowering time [48]. The literature discloses two major approaches to microbiome engineering: “synthetic communities” (SynComs) [49] and “host-mediated microbiome engineering” (HMME) [50], but they are still rarely applied to potato. The SynComs approach utilizes both synthetic biology and knowledge gained from microbial community analysis, metagenomics and bioinformatics. Coupled with a synthetic biology approach, this knowledge is valuable in the engineering of microbial consortia with predictable, stable and robust behavior [51]. HMME allows the establishment of long-term beneficial microbiome features by utilizing the host phenotype to iteratively select microbiomes indirectly. Its primary advantage over the SynComs approach lies in the majority of the chosen microbes being stress-resistant and closely related to the plant host. However, despite its elegance, this approach generally demonstrates modest efficacy, and the selection process can result in failure [52].

All types of potato microbiome studies should be complemented by the appropriate host studies, e.g., gene expression changes associated with pathogen invasion in tubers. Unfortunately, the number of transcriptome studies remains small and we suggest considering this area for further research. The existing approaches to studying the potato host plant and its microbiome are presented in the following scheme (Figure 1).

As mentioned above, there are no reviews dedicated to the potato microbiome. In the present review, we fill this gap by summarizing the factors that define the structure and diversity of the potato microbiome and by analyzing known inoculation experiments of potato plants. Our aim is to highlight the major findings from potato microbiome studies, propose promising methods of microbiome modulation and suggest areas for further research.

## 2. Survey Methodology

Google Scholar was used as the primary search engine. We searched for studies using the main keywords (“microbiome”, “potato”, “microbial community”) in various combinations with factors (“soil”, “drought”, “growth”, etc.), pathogen names (e.g., “Colorado beetle”) and species (e.g., *Ralstonia solanacearum*). To identify the articles containing sequencing data (amplicon or whole-genome), specific keywords were added (“DNA”, “sequencing”, etc.). Then, the relevant references and citations of the found papers were also analyzed to broaden the search field.

## 3. How Development Stages Define the Microbiome of Potato

The growth stage is the foremost factor to consider. As per various previous studies, the microbial community structure in the potato rhizosphere is mostly determined by the growth stage in comparison to cultivar type [21]. A three-year study conducted at a Dutch field site on the microbial community of various potato genotypes, sampled at different time points during plant development, revealed that specific bacterial genera were consistently present during the plant’s flowering stage [53,54].

When analyzing the effect of plants on the bacterial abundance and community structure, Özgül İnceoğlu et al. observed significant differences at all growth stages of potato plants [53,54]. An increase in abundance in the rhizosphere compared to the initial microbial composition in the soil was expected due to the presence of nutrient substrates released by plant roots through exudation, which is consistent with previous studies [11]. However, in some samples, the bacterial abundance remained roughly constant during the growing season. Possibly, the high organic and nutrient content of root exudates may have a significantly weaker effect on the bacterial abundance in this soil [54]. Moreover, the rhizosphere bacterial communities were also significantly different from the corresponding communities of the soil. This can be explained by the fact that the rhizosphere microbial communities are known to be influenced by complex interactions between soil type, plant species (genotype) and growth [55]. 

Similar observations were made by Qian Hou et al. when they studied the structural diversity of the rhizosphere microbiomes during potato cultivation in the field [56]. The microbial community of the rhizosphere changed significantly throughout the growth process. Microbiome diversity at the seedling stage differed strongly from other growth stages, but it did not substantially alter after the flowering phase, thus indicating a tendency to stabilization by the end of vegetation. This is consistent with previous studies that have shown a decrease in the rhizosphere populations as the plants mature [57]. Both the bacterial and fungal communities had significant differences at the sprouting stage and the first pre-sowing treatment of potato. They were stabilized at the flowering stage and then underwent relatively small changes until the potato crop matured [56]. The authors found that the relative abundance of Acidobacteriota and Candidatus Saccharimonadota in the bacterial community increased at the harvest stage compared to the seeding phase. At the seedling stage, the abundance of Bacteroidota increased and Chloroflexota significantly decreased. Meanwhile, in the fungal community, Ascomycota abundance increased (*p* < 0.05) at the seedling stage, while Basidiomycota abundance decreased (*p* < 0.05) [56]. In addition to specific taxa abundance, some authors consider the total abundance of microbes. Abdallah et al. revealed that the highest abundance of all culturable microorganisms was observed at seedling emergence, 15 days after harvest for late-maturing crops and at plant senescence for early-maturing varieties. Bacterial and fungal populations and actinomycetes were significantly increased by 35–55%, 14–18% and 17–42%, respectively, in the rhizosphere of all grown late-maturing potato varieties compared to the initial soil stage. The relative abundance of *Pseudomonas* spp., Actinomycetota, *Aspergillus* spp. and *Fusarium* spp. populations for all potato cultivars were 17.4%, 26–64%, 51–59% and 10–14% higher in the late season than in extra-early harvest season, respectively [58]. 

It can be concluded that the growth stage of potato plants is a crucial factor that significantly influences the microbial community structure in the rhizosphere. Several studies demonstrated that specific bacterial genera were consistently present during the flowering stage of potato plant development. The analysis of bacterial abundance and community structure at different growth stages revealed significant differences, with the expected increase in abundance in the rhizosphere due to nutrient substrates released by plant roots. Overall, understanding the dynamics of microbial communities in the potato rhizosphere at different growth stages is essential for optimizing potato plant health and production.

## 4. The role of Soil Structure and Moisture in Rhizosphere Microbiome Development

A key factor determining the composition of the rhizosphere bacterial community in potato plants is the soil. The tuber microbiota is largely recruited from the soil regardless of the potato variety and is transmitted from one generation to the next [9]. The investigations of the relationship between microbiome composition and soil type were constrained by a lack of large-scale research. Recently, a study of core microbiomes in several regions of the US (nine field sites total) was published by Klasek et al. [22]. The authors reported a substantial regional variation in the microbial communities, which was contrasted with consistency between seasonal and cultivar factors. Three major phyla were reported among the bacteria: Pseudomonadota, Acidobacteriota and Actinomycetota, which was consistent with previous studies cited above [10,21], but on lower layers of taxonomy the arising details complicated the overall picture [22]. Typically, the colonization of potato tubers from soil involves bacteria from the rhizosphere penetrating into the roots of potato plants, passing through the roots, and reaching the cortex or endoderm cell layer. Some bacteria can migrate through the xylem or intracellular spaces to the aboveground tissues of potato plants as well as to the stolon and subsequently into forming tubers [59]. In addition, bacteria colonizing mother tubers can migrate to the forming roots and further to other above- and belowground plant parts, including the stolon and subsequently, the next generation of tubers. For example, in this way the bacterium *Pectobacterium carotovorum* ssp. *atrosepticum*, which causes soft rot and blackleg disease, is transmitted from one generation of potato tubers to the next through the stolon [60]. 

For tuber-forming crops such as potato, the geocaulosphere, the thin zone of soil in contact with and influenced by the tuber, is a special habitat that exists between the potato and the soil environment. The geocaulosphere soil that remains associated with the tuber after harvest is called tare soil, and it also contributes to the overall composition of the rhizosphere community [61] in the next season. From harvesting to the dormancy period, the microbial community in tubers is subjected to major change, independent of the potato variety and the soil of origin. This indicates the dynamic nature of the potato tuber microbiota during storage [61,62]. The microbial communities in the tare soil of seed tubers can serve as a source of inoculum for the developing rhizosphere to promote biocontrol or growth. However, with the exception of potato pathogens, the microbes presented in tare soil are insufficiently studied, and there is also a lack of information on how non-pathogenic microbes in tare soil vary in composition or affect the plant health [63]. Most geocaulosphere studies are focused on fresh soils rather than on tare soils, which are part of the geocaulosphere and are associated with potato tubers after harvest [18]. The environmental conditions and storage time of seed tubers change the microbial abundance in tare soil, which results in a mismatch between the microbial communities of tare soil and fresh geocaulospheric soil [63].

While the initial potato plant rhizosphere community comes from seed tubers with tare and bulk soil, its further development is influenced by the soil structure, i.e., the composition of aggregates and degree of degradation. Numerous studies have shown that the soil bacterial diversity is higher in microaggregates than in macroaggregates [64], and the silt fraction has a more diverse bacterial community than the sand fraction [65]. Torsvik and Ovreas reported that 80% of the bacterial community was present in the micropores of stable soil microaggregates, suggesting that bacterial richness is related to microaggregation [66], but some bacteria may prefer other sizes of aggregates [67]. Soil compaction reduces soil porosity, water availability and the degree of soil aeration [64,68]. An associated shift in pore size distribution alters the microbial community structure, habitat and distribution and affects soil functions and ecosystem processes [69]. The soil structure also defines the evaporation character and ability to hold water, which is crucial for potato.

In agriculture, crop water consumption accounts for most of the total water use. It is expected that the demand for potatoes will continue to increase in the future, which will raise the need for irrigation. Potato is a water stress-sensitive crop due to its shallow rooting system [70,71]. In addition to improvements in irrigation management, it is necessary to understand the impact of soil water potential on the microbial community. Gumiere et al. evaluated the effect of four treatments of soil matrix potential (−15 kPa, −25 kPa, −30 kPa and −45 kPa) on the soil microbial community of three potato cultivars and two soil types (silt and sand). The results confirmed the potential of the soil matrix at the optimum irrigation level of −25 kPa, which promoted high potato production with minimum water consumption [72]. The irrigation levels affected the composition, predicted functionality and ecological network of the soil bacterial community. An excess (−15 kPa) and deficit (−30 and −45 kPa) of water increases microbial interaction and alpha diversity. The results showed a higher positive/negative ratio for Pseudomonadota, Actinomycetota and Acidobacteriota types at the optimum (−25 kPa) than at other irrigation levels.

The soil moisture coupled with macronutrient levels affect the microbial community. The influence of combined water and phosphorus limitation were studied in a field experiment dealing with the growth performance and plant characteristics of eight tetraploid and two diploid potato cultivars, as well as the diversity and functional potential of the root microbiome. Microbiome and metagenome analyses targeted the diversity and potential functions of prokaryotes, fungi, plasmids and bacteriophages and were related to plant characteristics such as tuber yield or crown closure time [73]. Different potato genotypes were found to respond differently to the combined stress and contained different microbiota in the rhizosphere and root endosphere. Proximity to the root, stress and potato genotype had a significant effect on the bacteria, while the fungal community was only slightly affected.

We can conclude that the microbial community in seed tubers undergoes significant changes during storage and can be transferred from one generation to the next with the influence of soil microbes. The diversity of the soil bacterial community is influenced by the composition of the aggregates and degree of degradation: it is higher in microaggregates compared to macroaggregates, and the silt fraction has a more diverse bacterial community than the sand fraction.

## 5. Impact of Bacterial and Fungal Pathogens on Potato Microbiome

Potato plants are vulnerable to a wide range of diseases related to the proliferation of pathogenic microorganisms. At the same time, some bacteria and fungi associated with the potato microbiome may demonstrate an antagonistic nature towards diverse pathogenic bacteria, fungi and oomycetes [74,75,76]. In addition, a number of potato symbiotic microorganisms are involved in the defense systems of plants and help them to resist pathogens by triggering the induced systemic resistance or systemic acquired resistance [77,78,79]. Subsequently, infection caused by a specific pathogen makes the potato more exposed to other threats as well. Thus, there is a complex process of interaction between the plant and pathogenic and endophytic microorganisms, and its mechanism remains largely unexplored.

Bacteria occupy a significant place among potato pathogens, especially in regions with warm and humid climates, such as tropical and subtropical areas [80]. The major bacterial diseases that pose a significant threat to potato crops are bacterial wilt [81], blackleg [59], soft rot [79] and common scab [2]. While bacterial wilt has a specific pathogen (*Ralstonia solanacearum*), soft rot in the tubers and blackleg in the stems are caused by a number of pectolytic bacteria belonging to the *Pectobacteriaceae* family: *Pectobacterium atrosepticum, P. carotovorum* subsp. *carotovorum* and *Dickeya* spp. [60,82,83,84,85]. Potato common scab (PCS) refers to soil-borne diseases that do not respond well to chemical control and require treatments that disrupt the integrity of the ecosystem [86]. The causative agents of PCS are the pathogenic *Streptomyces* spp. Interestingly, suppressiveness against PCS was also attributed to the genus *Streptomyces*, but to its non-pathogenic strains [87]. It should be noted that the mechanism of this inhibition of pathogenic *Streptomyces* is complex and, apparently, involves other bacterial species in addition to niche occupation by non-pathogenic *Streptomyces* bacteria. An effort to establish the relationship between the soil microbiome and PCS was performed by Shi and coauthors [15,88], described below.

The diversity and composition of the microorganisms in the four compartments of the soil–root system (geocaulosphere soil, rhizosphere soil, rooting layer soil and furrow soil) were studied on potato plants with high and low levels of common scab incidence [88]. Only in the geocaulospheric soil was it observed that there was a significant difference in microbiome between the high- and low-incidence groups. In contrast to previous studies, the low-disease group showed higher diversity and higher complexity of the co-currency network. *Variovorax*, *Stenotrophomonas* and *Agrobacterium* were the most abundant genera that were significantly and positively correlated with the level of common scab severity and the abundance of pathogenic *Streptomyces*. In contrast, *Geobacillus*, *Curtobacterium* and unclassified *Geodermatophilaceae* were negatively correlated with these two parameters. In another study, potato common scab did not cause significant differences in the composition of endophytes associated with roots, tubers and stems between high- and low-infestation groups at the community level, but impacted the relative abundance of a few specific endophytes [15].

The impacts of diseases on the community structure of the potato microbiome have not yet been comprehensively researched. Rasche et al. compared the endophytic community shifts for the different potato lines (transgenic and conventional) with or without treatment by the *Pectobacterium carotovorum* ssp. *atrosepticum*. Endophytic populations of different potato lines were found to respond distinctly to bacterial invasion [89]. However, a clear effect of pathogen impact on the microbial community structure was found with respect to the soil and growth stages of the plants. Another study where potatoes of several lines were also infected with the bacterium *Pectobacterium carotovorum* ssp. *atrosepticum* demonstrated that the bacterial diversity increased in the infected plants compared to control ones [90]. The authors attributed the increased diversity of endophytes in contaminated potatoes to their defensive function against the pathogen. For instance, 38% of endophyte isolates showed antagonistic activity to the blackleg causative agent.

Another study on soft rot and blackleg disease tested the correlation between the degree of disease severity in tubers and differences in the microbiome structure of tubers and soil [79]. According to the results of the study, there were significant differences (PERMANOVA analysis) between the bacterial and fungal communities in tubers and soil with high and low disease lesions. In both tuber and soil samples with high incidence, higher numbers of *Bacteroidetes* were found, while in samples with low incidence more Actinomycetota and Bacillota were observed. As for fungi, the Ascomycota phylum was the most abundant in all the treatments. An abundance analysis in DESeq2 showed that 310 bacterial OTUs differed in abundance between high and low disease incidence in tubers. Representatives of the genera *Staphylococcus, Rhodococcus*, *Pseudomonas*, *Pantoea*, *Curtobacterium* and *Arthrobacter* had higher abundance in tubers with low disease incidence. The random forest analysis also confirmed that the genera *Pantoea*, *Arthrobacter* and *Rhodococcus* best explained the differentiation between lots with low and high disease activity. Regarding the fungi, 41 OTUs showed differences in abundance between tuber lots with high and low disease incidence. Representatives of the genera *Vishniacozyma*, *Pyrenochaeta* and *Acremonium* were the most abundant in several low-disease-affected lots. For the soil samples, 826 bacterial and 277 fungi OTUs were identified as significantly differentially abundant. The bacterial genera *Terrabacter*, *Sphingomonas*, *Bacillus* and *Bradyrhizobium* and the fungal genera *Vishniacozyma*, *Fusarium*, *Cladosporium*, *Trichoderma, Pyrenochaeta* and *Acremonium* demonstrated higher abundance at low disease incidence.

In turn, Mao et al. noted that in the rhizosphere microbiome of potatoes infected with blackleg, *Flavobacterium*, *Acinetobacter*, *Dickeya*, *Sphingobacterium* and *Myroides* were predominant, while *Bacillus*, *Rhodoplanes*, *Pedobacter* and an unidentified genus from *Gaiellaceae* were less abundant compared to the microbiome of healthy potatoes [91]. In a follow-up study, the authors extended and updated their results. The investigated potato diseases also included bacterial wilt. The authors showed that in the blackleg disease group, the increase in bacterial abundance was mainly attributed to Bacillota such as *Paenibacillus*, *Sporosarcina* and *Caloramator*, while in the bacterial wilt group it was mainly attributed to Betaproteobacteria such as *Novosphingobium* and *Rubrivivax* [92]. The influence of the season and geography of the experiments was considered to be a probable reason for the discrepancy in the results.

## 6. Relationship between the Potato Microbial Community and Pests

The major yield losses for potato plants are caused by pests (e.g., leaf miners, aphids and the Colorado potato beetle) [5,93]. It was confirmed that the rhizosphere microbial communities affected plant defenses against herbivore attacks [94]. Although all aspects of this interaction are still not fully understood, some mechanisms have been investigated in detail. One of the most studied ways in which microbial communities influence plant–insect interactions consists in the alteration of plant signaling and defense mechanisms. This way involves the ability of soil mutualists to regulate the induction of plant phytohormones such as jasmonic acid, salicylic acid and ethylene [95,96]. Exposure to plant hormone signaling systems can lead to gene expression and biosynthesis of secondary metabolites, plant defense proteins, enzymes and volatile organic compounds (VOCs) [97]. It is still a crucial research challenge to identify the microbiome members responsible for inducing plant defenses. Based on previous research, it is possible to distinguish the contribution of the following genera: *Azospirillum*, *Arthrobacter*, *Acinetobacter*, *Bradyrhizobium*, *Pseudomonas*, *Brucella*, *Glutamicibacter*, *Bacillus*, *Erwinia*, *Ralstonia* and *Rhizobium* [95,98].

The study of herbivory’s effect on the rhizosphere bacterial microbiota in potato plants highlights another side of the three-way interactions [99]. It not only accounts for the influence of herbivores on the plant microbiome, but also analyzes how herbivore types (aphids, nematodes and slugs) differ in their influence on the rhizosphere bacterial microbiota of potato. As a result, the investigation confirms previous findings that the diversity and structure of the rhizosphere bacterial microbiome are altered by herbivores [100,101]. Some authors suggested that plants attract beneficial microbes to their roots in response to insect attack, including by changing the chemistry of root exudates. [102,103]. However, the assumption that various types of herbivores affected microbiome diversity and structure differently was not supported by the preliminary results. Nevertheless, a deeper analysis showed that herbivore type influenced the structure of the microbial co-occurrence network.

Regarding potato specificity, a correlation between bacterial genera and nematode species (*Pratylenchus neglectus* and *Meloidogyne chitwoodi*) was determined, with all samples being taken from five different Colorado potato farms [104]. Some nematode species act as worldwide plant parasites that can cause serious damage to many important crops, including potatoes. Castillo et al. found that the abundance of *Bacillus* spp., *Arthrobacter* spp. and *Lysobacter* spp. in potato soil was negatively correlated with the abundance of *P. neglectus* and *M. chitwoodi*. It was suggested that these three genera were antagonists of plant-parasitic nematodes by producing different compounds.

Gao et al. revealed that the S2 strain of *Bacillus cereus* exhibited high nematicidal activity and produced some extracellular substances to kill nematodes [105]. Treatment of nematodes with its culture resulted in the killing of 77.89% of *Caenorhabditis elegans* and 90.96% of *M*. *incognita*. It was also shown that *Bacillus pumilus* L1 could be used against the root-knot nematode, *Meloidogyne arenaria* [106].

Research on the interaction between potato and pest microbiomes gave remarkable results [107]. Thus, potato endophytes protected plants not by directly harming insects with toxins, for example, but by negatively affecting insect microsymbionts. The endophytic strain *Bacillus subtilis* 26D was found to increase Colorado potato beetle mortality by disrupting the insect symbionts *Enterobacter* ssp. and *Acinetobacter* ssp.

A three-way interaction between potato, its microbiome, and insects may also occur when insects carry some types of bacteria, thereby transmitting them to the plant’s microbiome. For example, the potato psyllid (*Bactericera cockerelli*), a pest of solanaceous crops including potatoes, transferred the phytopathogen “*Candidatus* Liberibacter solanacearum” [108]. This bacterium is recognized as the causative agent of zebra chip in potatoes. Rossmann et al. concluded that pathogenic soft rot Enterobacteriaceae (SRE) causing diseases in potato was a natural member of some insect microbiomes, including *Delia floralis* carrying more SRE than *Plutella xylostella* and carnivorous green lacewing larvae [109].

As a result, to obtain a sufficiently complete picture of the interaction between the potato microbiome, the plant and the pests to which it is exposed, it is necessary to consider completely different perspectives and mechanisms of their relationship. In addition, it is very likely that there are other interaction mechanisms hidden from researchers at the moment that have yet to be discovered. However, it is indisputable that the microbiome can have a significant impact both on the condition of the plant, on its ability to protect itself, and on the insects that interact with this plant, including an impact on the microbiome of these insects.

## 7. Potato Microbiome Modulation

Extensive research was devoted to rhizobacteria and AMF that promote plant growth, mitigate disease impact and reduce the amendment usage. However, most of the studies were not focused on particular plant species or families [110]. An effective strategy to face the detrimental effects of environmental factors is to inoculate potato plant tubers with various types of microbial mixtures or add them to the soil (Figure 2). This section gives a concise overview of some recently successful cases that were specifically aimed at potato. Unfortunately, only a few studies considered changes in the microbiome induced by inoculation, and none of them employed the HMME or SynComs approaches mentioned above. The studies involving AMF inoculation commonly focus on increasing the yield, neglecting the microbiome change [111]. For the convenience of the reader, we have summarized the mixture components, effect observed and raw data references from these studies in Table 1.

The most common species for inoculation of potato plants belong to the *Pseudomonas* and *Bacillus* taxa. Both are widely presented in bulk soil and the rhizosphere [17,112,113], and have a diverse metabolic arsenal to promote plant growth and protect against pathogens. Beneficial microbes could help the potato plant acquire the important potassium (K), calcium (Ca), and magnesium (Mg) metal ions that are required in relatively large amounts (Figure 2) [114]. Among seven key soil macroelements, K is one of the most abundant and necessary for normal potato plant nutrition due to its key role in starch synthesis, quantity and quality of tubers [115]. Representatives of the genus *Bacillus* were reported to possess the ability to solubilize K for tea plants [116]. A similar approach was applied to potatoes growing in salt-affected soils. Tahir et al. reported that indole-3-acetic acid-producing bacteria (*Bacillus* sp. SR-2-1/1) helped to balance ion uptake in potato plants. These bacterial strains could be used solely or in a mixture with other microorganisms as bioinoculants [117]. Another key microelement, phosphorus (P), could be a limiting factor in potato plant growth and decreased photosynthetic rate, particularly in the case of declining soil fertility [118]. To enhance the acquisition of P, Pantigoso et al. used a phosphorus-solubilizing bacteria isolated from the wild potato (*Solanum bulbocastanum*) rhizosphere [113]. The authors identified three P-solubilizing bacterial strains as *Enterobacter cloacae*, *Pseudomonas oleovorans* and *Bacillus thuringiensis*. Isolated strains could assimilate organic and inorganic P and promote potato growth and yield, but the efficacy of such a bacterial mixture has to be evaluated in the field.

Unfortunately, good nutrition alone is insufficient for high yield of potato plants. As discussed above in Section 5, diseases of a bacterial nature are an important factor in reducing yield and storage time. *Pectobacterium carotovorum* subsp. *carotovorum* and *Pectobacterium atrosepticum* bacteria cause blackleg and soft rot diseases [119], mentioned above. A promising approach to control this disease is to isolate the bacteria with antagonistic activity (Figure 2). Padilla-Galvez et al. identified native Chilean potatoes as an unexplored source of endophytes and isolated *Streptomyces* sp. TP199 and *Streptomyces* sp. A2R31, which were able to inhibit the growth of the pathogens. Another widespread bacterial disease is potato common scab (PCS), and there are a number of efforts to apply biofertilizers as biocontrol agents of such diseases. For example, phenazine-1-carboxylic-acid (PCA) production by *Pseudomonas fluorescens* LBUM223 altered the expression of key virulence genes in *Streptomyces scabiei*. The authors highlighted the autochthonous soil microbial community, which was sustainable to inoculant bacteria like *Pseudomonas*. They reported that in both cases (single or multiple inoculation with *Pseudomonas fluorescens* LBUM223) the geocaulosphere microbial community did not change significantly, but when applied biweekly, the concentration of LBUM223 remained stable (about 10^7^ bacteria/g of soil). It was concluded that the application of the biofertilizer *Pseudomonas fluorescens* LBUM223 was safe (not disturbing the soil community) and productive (increasing potato yield and controlling common scab) [17]. In a further review, Biessy and Filion described the diversity of the causing agent (*Streptomyces* spp.). They stressed that genomic information of biofertilizer strains is the key to understanding the mechanism of biocontrol of the disease. While many species of biocontrol bacteria were studied, the relationship between their genomic features and the level of *Streptomyces* suppression is far from clear. The promising *Pseudomonas* strain LBUM223 had phenazine-producing potential and was reported to be effective against multiple *Streptomyces* species [2]. A similar study showed strong correlations between the biosynthesis of specialized metabolites by a population of *Pseudomonas fluorescens* and the antagonism of the bacterial and oomycete pathogens *Streptomyces scabiei* and *Phytophthora infestans*, respectively [120].

The lesions of PCS are frequently reduced using a microbial mixture (consortium) of bacteria and fungi. Wang et al. applied rhizosphere-derived microbial products (*Bacillus subtilis* strain znjdf1 and *Trichoderma harzianum* strain znlkhc1). The control and treatment samples differed only in the amount of microbial product used. The severity of the disease was significantly lowered in the selected fields for six years: the PCS disease index decreased from 63–69% to 46.1% (2016) and 31.7% (2017) [121].

In Section 5, the study of Shi et al. was already mentioned, which reported that the geocaulosphere had the most important role in the severity of common scab: the composition and functional gene content of the microbial community were strongly associated with the direct influence of pathogenic organisms (mostly *Streptomyces*). Bacteria of the *Bacillus* and *Pseudomonas* taxa could inhibit the ThxA (scab phytotoxin) biosynthesis and reduce the *Streptomyces* population [88]. A relatively novel approach to suppress the common scab is the application of non-pathogenic strains of *Streptomyces*. Hiltunen et al. compared the structure of the microbial community in the geocaulosphere (the authors called it the tuberosphere) in treatment and control cases. The authors showed that the treatment of tubers with the *Streptomyces* strain (Str272), when applied systematically, enhanced the microbial diversity, and the soil preserved the ability to suppress the diseases for several seasons [87]. The study was limited only to the bacterial community impact, but many other factors (e.g., chemical soil properties) could be important too.

Apart from the conventional approach of methodically isolating strains for subsequent individual or combined application as inoculants, future research could be based on the ability of plants to selectively attract specific microorganisms when confronted with a particular pathogen. These attracted microorganisms could then form synthetic communities and their potential for protection against the initial pathogen could be assessed (as well as other desired characteristics). This developing area of microbiome management is expected to present novel options for safeguarding plants from diseases [112]. To test this approach, De Vrieze et al. applied nine potato-associated *Pseudomonas* strains in various combinations (in comparison with sole-strain cultures). The authors demonstrated the potential benefits of combining compatible strains and found that a dual mixture provided stronger and more consistent protection than the single strains. The complexity of interactions even in the limited consortia remained hard to untangle [112].

In addition to bacterial diseases, potato suffers from fungal pathogens. The well-known Verticillium wilt that is caused by the soil-borne pathogen *Verticillium dahliae* can be controlled by inoculation with particular bacterial species. Song et al. showed that the *Bacillus subtilis* strain Bv17 could penetrate into the potato rhizosphere and significantly decreased the propagation and expansion of *Verticillium dahliae*. However, the underlying functional mechanisms of these bacteria’s action remain unknown [122]. The authors report that the diversity of the soil microbiome was significantly different in treated and control samples of the soil. Unfortunately, the raw data for this study were not published. Another devastating disease of potato and tomato is late blight (caused by *Phytophthora infestans*, which is an oomycete). *Bacillus subtilis* is one of the most commonly used biocontrol agents for various potato diseases. For example, its application together with chitosan conjugates with caffeic and ferulic acids resulted in increased resistance of potato to the late blight pathogen [123].

The arsenal to face bacterial and fungal pathogens could be enriched by phages. A relatively rare approach to controlling soft rot involves the application of bacteriophages. The positive aspects of phages, such as host specificity, ecological soundness, self-replication, non-toxicity and the ability to overcome antimicrobial resistance, have sparked a growing interest in their use for the biocontrol of plant pathogens. Mousa et al. applied a mixture of three phages that were known to be effective against soft rot-causing bacteria (mostly belonging to the genus *Pectobacterium*). The authors observed changes in the rhizosphere microbiota, which confirmed the significant role of bacteriophages in shaping the microbiome related to potatoes in the rhizosphere. The improvement in plant health following the application of various phages may not be solely restricted to reduced pathogen impact, but also due to changes in the composition of microorganisms [81]. It was also demonstrated that the application of fungal agents and some bacterial species can be an effective and non-toxic remedy in the control of soft rot disease [124].

As mentioned in the Introduction, microbiome engineering applications (including the SysComs and HMME approaches) have not yet been described for potato. However, new experiments and methods could be suggested basing on the results for other crops. Due to the potato’s susceptibility to drought, we have provided some relevant examples below. Jacquiod et al. hypothesized that artificial selection of microbiota could be successful if the community structure had been stabilized previously [125]. The authors conducted a large-scale experiment with 1800 plants of *Brachypodium distachyon*. They specifically chose rhizosphere microbiota associated with high/low leaf greenness (related to plant performance). The study results showed a significant correlation between the variability in plant features and specific microbiota structures. Two distinct sub-communities were identified in relation to high or low leaf greenness. The abundance of these sub-communities was influenced through directional selection. In another study, Jochum et al. first used HMME to select beneficial microbial communities that promoted wheat (*Triticum aestivum* L.) plant tolerance to drought stress [50]. The ability of the wheat to survive drought was increased through the six rounds of the experiment (HMME selection). The effect on plant water stress tolerance was transferred by the inoculum. Autoclaving the HMME inoculum eliminated the effect on seedling water deficit tolerance. The experimental design was challenging due to the impractical separation of soil components with microbes entirely. These findings suggested that the alteration in plant adaptation to drought stress was tightly related to microbial population dynamics [50].

The abovementioned successful examples of microbiome modulation of potato plants provide valuable knowledge. It should be noted that some studies lack metagenomic or amplicon sequencing data to allow replication of the results, ecological studies or meta-analysis. While the number of successful cases described above is certainly not exhaustive, it does allow us to define future research areas in Section 7.

**Table 1 ijms-25-00750-t001:** Potato inoculation studies summary.

Microbial Mixture Components	Effect after Inoculation	NCBI BioProject	Reference
*Rhizophagus irregularis*, *Funneliformis mosseae*, *Claroideoglumus etunicatum*	increase both the yield and nutritional quality of potatoes	n/a	[111]
*Enterobacter cloacae, Bacillus thuringiensis,* and *Pseudomonas pseudoalcaligenes*	increase biomass, yield and P nutrient uptake in potato plant	n/a	[113]
Nine *Pseudomonas* strains	dual mixture provided stronger protection against *Phytophthora infestans* than the single strains	n/a	[112]
*Streptomyces* sp. TP199 and *Streptomyces* sp. A2R31	applied strains possess antagonistic activity in vitro against *Pectobacterium carotovorum* subsp. *carotovorum* and *Pectobacterium atrosepticum*	n/a	[126]
*Pseudomonas fluorescens* strain LBUM223	LBUM223 capable of controling PCS while not disturbing autochthonous microbiome	PRJNA436092	[17]
*Pseudomonas fluorescens*	metabolite biosynthesis correlates with antagonism of the potato pathogens *Streptomyces scabies* and *Phytophthora infestans*	PRJEB34261	[120]
*Bacillus subtilis* strain znjdf1 and *Trichoderma harzianum* strain znlkhc1	applied strains suppress PCS and increase tuber yield	PRJNA512875	[121]
*Bacillus subtilis* strain Bv17	*Bacillus subtilis* strain Bv17 treatment was able to decrease diseases of potato and improve the quality and quantity of yield	n/a	[122]
*Streptomyces* (various)	non-pathogenic *Streptomyces* reduce the population of pathogenic *Streptomyces* due to niche overlap	PRJNA477767	[88]
*Streptomyces* strain (Str272)	*Streptomyces* strain Str272 has antagonistic activity against PCS	PRJEB40435	[87]
Phages PSG11, WC4 and CX5	plant health was enhanced after phage application, probably due to pathogen elimination and shifts in the microbiome composition	PRJNA867554	[81]

## 8. Conclusions and Perspective

There are many factors influencing the potato microbiome, which can be divided into two parts: “core”, the most common microorganisms, and “satellite”, whose structure is subjected to more significant changes under the influence of external factors. The main source of microorganisms for the potato microbiome, regardless of variety, is the soil, as well as those fungi and bacteria that are contained in the seed tubers.

As the plant grows, the microbiome generally remains quite stable, especially in the post-flowering stages, but may undergo changes under the influence of both abiotic and biotic factors discussed in our review. The most dangerous potato diseases have a bacterial (wilt, blackleg and soft rot symptoms and common scab) or oomycetal (late blight, early blight, black scurf, dry rots, silver scurf, wart and charcoal rots) nature. It is important to distinguish pathogenic and non-pathogenic strains (e.g., *Streptomyces*) and use the latter to control the diseases. The complex mechanism of pathogen inhibition by the non-pathogenic strain apparently involves other bacterial species. It is worth noting that incorporating metabolomic and proteomic data with metagenomic analysis could lead to a more profound understanding of this mechanism. Interestingly, in most studies, microbiome diversity was lower in healthy than in infected plants. It was suggested that microorganisms contributing to increasing microbiome diversity in diseased plants play a protective role against pathogens, which is consistent with the “cry for help” hypothesis, but it is important to evaluate the ability of potato plants to recruit beneficial microorganisms.

Less obvious, but significant, is the role of insects in the development of the potato microbiome. For example, insects are carriers of various bacteria, fungi, oomycetes and protozoa, including pathogenic species. In response to insect attack, plants produce various substances, regulated by the activity of the endophytic microbiome of the potato, that are part of their defense system. Some species of the potato microbiome may serve as direct antagonists to insects by producing various compounds or by disrupting the insect microbiome to cause the death of the insect itself.

Inoculation and microbial engineering are actively used to overcome the negative effects mentioned above, accelerate growth and increase potato yield. It allows significant improvements to be achieved without contaminating the soil with toxic substances that are destructive to beneficial microorganisms and insects. The most known and widely used biocontrol agents belong to the *Pseudomonas* and *Bacillus* taxa, but microbiome engineering approaches have not yet been described for potato. Drought- and pathogen-resistant potato varieties could be sources of microorganisms for further inoculation. While endophytic bacteria are commonly used as inoculants, a high-throughput cultivation pipeline has to be developed for potato plants. Using a microbial mixture rather than a monoculture represents a more promising approach due to the greater stability and adaptability of the resulting community. We suggest paying more attention to the metabolic potential of biofertilizers (inoculants) in addition to the microbiome structure. Moreover, the inoculated microorganisms, particularly in synthetic communities, may displace the autochthonous species, which could disrupt the metabolic networks in the microbiome. In addition to whole-genome data, the gene network approach could disentangle some complex plant–microbe interactions.

All this allows us to highlight the great role of the microbiome in the functioning and development of potato and its defence mechanisms when faced with negative impacts. With proper study, such protective microbiome-associated mechanisms can be artificially regulated. We summarized the major findings of the potato plant microbiome, covered in the present review in Table 2. In order to achieve a better reproducibility and an easy comparison of the results, we recommend the development of a protocol for potato plant microbiome studies where the irrigation regime, physico-chemical soil properties, sample preparation and DNA extraction technique could be standardized. To employ the full potential of the potato microbiome, future studies could use a combined approach including microbiology, metagenomics, metatranscriptomics and metabolomics methods.

## Figures and Tables

**Figure 1 ijms-25-00750-f001:**
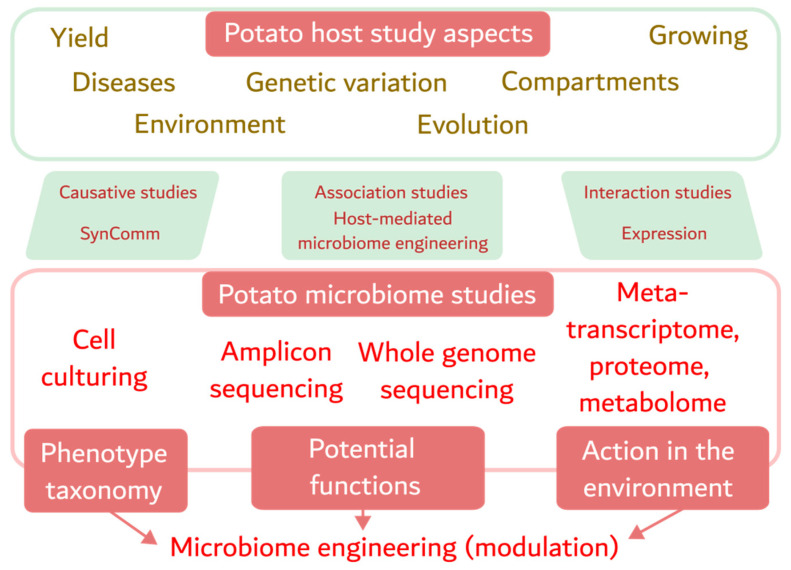
The proposed scheme shows the role of microbiome engineering in potato research. The structure relates the main objects of potato studies to three types of research: causative, associative and interactional. It also describes the current methodology and the subjects it addresses: phenotype taxonomy, potential functions and interactions with the environment.

**Figure 2 ijms-25-00750-f002:**
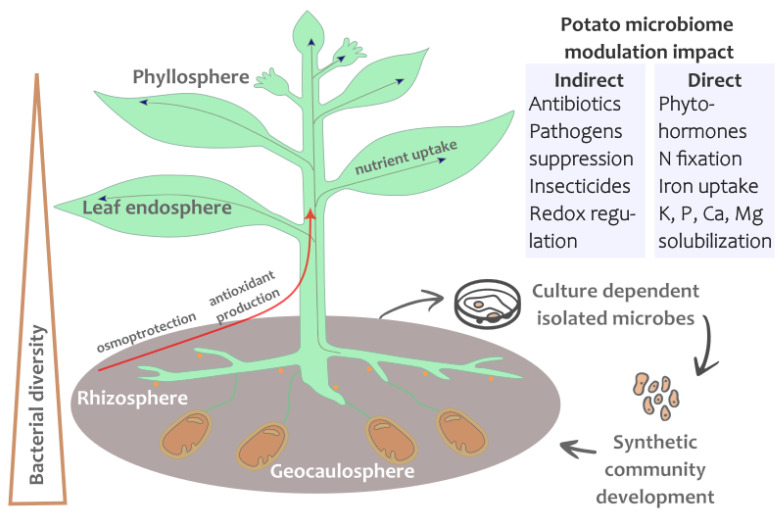
Potato plant microbiome impact. The composition of the microbiome depends on the part of the plant: compartments are rhizosphere (near root soil), geocaulosphere, endosphere (inner tissue of the plant) and phyllosphere (leaves and stems surface). Microbial inoculant can promote plant growth by direct and indirect mechanisms. Rhizosphere is often a source of microbes, the cultures of which could be used to produce the inoculants (microbial mixtures). Developed synthetic communities from the soil could be inoculated into another soil microbiome to suppress soil pathogens and pests.

**Table 2 ijms-25-00750-t002:** Summary of the major findings of the potato plant microbiome.

Finding Scope	Description	Reference
Core microbiome taxa	The main representatives of the potato core microbiome are *Bradyrhizobium*, *Sphingobium* and *Microvirga*; the most abundant genera in the rhizosphere are *Lentzea* and *Streptomyces*	[10,21,22,23]
Growth stages	Specific bacterial genera were consistently present during the flowering stage of potato plant development	[53,54]
Soil type influence	The diversity of the soil bacterial community is higher in microaggregates and the silt fraction than in macroaggregates and the sand fraction	[64,65,66]
Native potato isolates	*Streptomyces* sp. TP199 and *Streptomyces* sp. A2R31 could inhibit the growth of *Pectobacterium carotovorum* subsp. *carotovorum* and *Pectobacterium atrosepticum*	[126]
Non-pathogenic bacterial strains	The are multiple cases of biocontrol of PCS and other diseases using non-pathogenic bacterial strains of *Baccilus*, *Pseudomonas* and *Streptomyces* as antagonists to pathogenic ones	[17,81,87,121,122]
Occurrence of nematodes influence on microbiome	Abundance of *Bacillus* spp., *Arthrobacter* spp. and *Lysobacter* spp. in potato soil was negatively correlated with the abundance of *P. neglectus* and *M. chitwoodi* due to parasitic nematodes antagonism	[104]

## Data Availability

No new data were created or analyzed in this study. Data sharing is not applicable to this article.

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
