# Peer review of "Potato Microbiome: Relationship with Environmental Factors and Approaches for Microbiome Modulation"

_ijms, 2024, doi:10.3390/ijms25020750_

Round 1

Reviewer 1 Report

Comments and Suggestions for Authors

The manuscript submitted to my review, entitled: “Potato microbiome: relationship with environment factors and approaches for microbiome modulation” is a review study on the microbiome of Solanum tuberosum. The work is divided into seven subchapters, including Introduction, five chapters on various aspects of research on the potato microbiome and Conclusions.

Research on the soil microbiome is an important and current topic that can certainly gain readers' attention. The idea of synthetically collecting scientific studies on one species, which is a very important, even key crop in some regions of the world, is a good idea. The work raises a number of important issues regarding microbiome research.

Preparing a good review work is a difficult task, because works of this type are expected not only to indicate the results of previously published works, but also to take a broad, mature and critical look at the discussed issue. In the case of the manuscript being assessed, the Authors should work intensively on the style and the so-called flow of the text, because in its current form the text is difficult to follow. One paragraph should be consistent with the previous one, or result from the previous one. In present form the threads seem detached from each other. To better guide the reader through individual issues, I would recommend rewording each subchapter, starting from general information to specific ones, and more focusing on particular threads than on separate studies one by one.

Citations should also be supplemented in many places, especially in the Introduction. You can’t write that there is a lot of research on a given issue and not cite some of the most important works in this sentence.

I also suggest expanding the manuscript with a chapter on perspectives in potato microbiome research. This could be included as the penultimate chapter.

The methodology of conducting research on the microbiome is also a very interesting issue and it must be remembered that it has an impact on the results obtained. It is worth analyzing the cited works in terms of the methods used - were the bacteria determined using microbiological methods or DNA sequencing? If so, what sequencing method was used – using PCR amplification or analysis of the native metagenome by nanopore sequencing? This is of great importance in estimating bacterial abundance.

Finally, please, correct numerous typos in the text.

Author Response

Dear Editor and Reviewers!

We thank you for your generous comments on the manuscript and have revised the manuscript according to your suggestions. In this version, all comments were addressed point-by-point and revisions were made. We hope our revisions are satisfactory to you and the reviewers.

The original manuscript was revised to address the topic of potato microbiome research, emphasizing the relationship with environmental factors and approaches to microbiome modulation. We found no previous reviews dedicated to potato microbiome modulation or related topics. Therefore, the aim of our review is to summarize recent studies on this topic across spectrum of factors (drought, pathogens, soil, etc.)

Our current project focuses on the modulation of the potato microbiome by the biofertilizer Rhodococcus qingshengii VKM Ac-2784D (https://rscf.ru/project/23-26-10049/). The introduction has been revised to more clearly connect the body sections and to show the interrelation of their results with the main objective of review. A figure has been added to the manuscript to demonstrate the relationship between the potato microbiome impact factors discussed in the sections and the practices of microbiome engineering. The conclusion was supplemented by findings and perspectives concerning the development of potato microbiome research. All sections of the manuscript were expanded with proper references according to reviewers’ comments.

Reviewer 1

Q: - 1: The Authors should work intensively on the style and the so-called flow of the text, because in its current form the text is difficult to follow. One paragraph should be consistent with the previous one, or result from the previous one. In present form the threads seem detached from each other. To better guide the reader through individual issues, I would recommend rewording each subchapter, starting from general information to specific ones, and more focusing on particular threads than on separate studies one by one.

A: We have revised the introduction and conclusion to emphasize the main purpose of the study and to show the relationship between the theme and results of the paragraphs and the key aspects of potato microbiome studies and engineering. A figure has been added to make this relationship clearer. The paragraphs have also been modified to add a more logical and coherent flow of the text. The conclusion section was extended with perspectives.

Q: - 2: Citations should also be supplemented in many places, especially in the Introduction. You can’t write that there is a lot of research on a given issue and not cite some of the most important works in this sentence.

A: Citations have been added in all sections, including in the Introduction, to support the issues stated.

Q: - 3: I also suggest expanding the manuscript with a chapter on perspectives in potato microbiome research. This could be included as the penultimate chapter.

A: Perspectives on potato microbiome research are reviewed and expanded in the conclusion of the manuscript.

Q: - 4: The methodology of conducting research on the microbiome is also a very interesting issue and it must be remembered that it has an impact on the results obtained. It is worth analyzing the cited works in terms of the methods used - were the bacteria determined using microbiological methods or DNA sequencing? If so, what sequencing method was used – using PCR amplification or analysis of the native metagenome by nanopore sequencing? This is of great importance in estimating bacterial abundance.

A: We agree with this suggestion, but the analysis of the methods is beyond the scope of our review and requires a completely new manuscript. For future review we will consider such an approach for microbiome engineering studies.

Q: - 5: Finally, please, correct numerous typos in the text.

A: The typos, style errors and English language have been corrected.

Reviewer 2 Report

Comments and Suggestions for Authors

General remarks

The data presented are interesting and important, but in my opinion the review is a bit too „shalow”. I think it could be broader, deeper, more elaborated. The data you have collected should lead to useful conclusion (to something new and original), while conclusion in this manuscript rather confirm well known information. Moreover, you  should underline more clearly what was the reason for taking under consideration this topic.

Some other suggestions for improving are below:

Line 28 – 29 „Particular attention is paid to the following pests: nematodes [4], potato tuber moth [5], aphids [6], and leaf miner [7]” – what about Colorado potato beatle (Leptinotarsa decemlineata)– the most important pest of potato plants?

Line 38, 176 – add reference

Line 46, 57 – lack of dot

Lines 51 – 52 – giving latin name is necessary only once – at the first mentioning – you have mentioned Solanum tuberosum in line 25.

Line 52 – lack of space between „[14,15].” and next sentence, similar errors in lines 183, 344

 Line 81, 85,126, 146, 317, 328 and in many other places in the manuscript  – unnecessary space

Line 82 – „is” two times

After Introduction there should be short paragraph about methods description (used key words, search engines, etc.)

Line 172 – „Most geocaulosphere studies have focused on fresh soils rather than on tare soils, …..”- add these references

 Line 243 – Sometimes the same latin names (here Streptomyces) are written differently – I mean one time in italics, other time with regular font – please check the whole manuscript and correct.

 Line 423 – should be „application”

Line452 – should be „fungal”

Comments on the Quality of English Language

English fine

Author Response

Dear Editor and Reviewers!

The original manuscript was revised to address the topic of potato microbiome research, emphasizing the relationship with environmental factors and approaches to microbiome modulation. We found no previous reviews dedicated to potato microbiome modulation or related topics. Therefore, the aim of our review is to summarize recent studies on this topic across spectrum of factors (drought, pathogens, soil, etc.)

Our current project focuses on the modulation of the potato microbiome by the biofertilizer Rhodococcus qingshengii VKM Ac-2784D (https://rscf.ru/project/23-26-10049/). The introduction has been revised to more clearly connect the body sections and to show the interrelation of their results with the main objective of review. A figure has been added to the manuscript to demonstrate the relationship between the potato microbiome impact factors discussed in the sections and the practices of microbiome engineering. The conclusion was supplemented by findings and perspectives concerning the development of potato microbiome research. All sections of the manuscript were expanded with proper references according to reviewers’ comments.

Reviewer 2

Q: - 1: The data presented are interesting and important, but in my opinion the review is a bit too „shalow”. I think it could be broader, deeper, more elaborated. The data you have collected should lead to useful conclusion (to something new and original), while conclusion in this manuscript rather confirm well known information. Moreover, you should underline more clearly what was the reason for taking under consideration this topic.

A: We agree, that there are different ways to organize the review, but depth requires narrowing the topic. We have revised the introduction and conclusion to emphasize the main purpose of the study and to show the relationship between the theme and results of the paragraphs and the key aspects of potato microbiome studies and engineering. A figure has been added to make this relationship more visual. Additional cases have been added to the last section to show the most current and relevant microbial engineering techniques as an answer to perennial problems in potato cultivation. The concluding part of the manuscript is enriched with new insights and perspectives for the development of potato microbiome research. The motivation of writing the review described in Introduction section.

Q: - 2: Line 28 – 29 „Particular attention is paid to the following pests: nematodes [4], potato tuber moth [5], aphids [6], and leaf miner [7]” – what about Colorado potato beatle (Leptinotarsa decemlineata)– the most important pest of potato plants?

A: In section 5 we consider the Colorado beetle, but missed it in Introduction. Colorado potato beatle (Leptinotarsa decemlineata) has been added to the list of pests with proper citation.

Q: - 3: Line 38, 176 – add reference

Line 46, 57 – lack of dot

Lines 51 – 52 – giving latin name is necessary only once – at the first mentioning – you have mentioned Solanum tuberosum in line 25.

Line 52 – lack of space between „[14,15].” and next sentence, similar errors in lines 183, 344

 Line 81, 85,126, 146, 317, 328 and in many other places in the manuscript  – unnecessary space

Line 82 – „is” two times

Line 172 – „Most geocaulosphere studies have focused on fresh soils rather than on tare soils, …..”- add these references

 Line 423 – should be „application”

Line452 – should be „fungal”

A: Appropriate references have added and mistakes have been corrected.

Q: - 4: After Introduction there should be short paragraph about methods description (used key words, search engines, etc.)

A: We added the special section 2 “Survey Methodology”

Q: - 5: Line 243 – Sometimes the same latin names (here Streptomyces) are written differently – I mean one time in italics, other time with regular font – please check the whole manuscript and correct.

A: The latin names have been corrected according to the International Code of Nomenclature of Prokaryotes. Prokaryotic Code (2022 Revision) (Oren A, Arahal DR, Göker M, Moore ERB, Rossello-Mora R, et al, 2023; https://www.microbiologyresearch.org/content/journal/ijsem/10.1099/ijsem.0.000778?crawler=true)

Moreover, we see the papers in IJMS with (https://www.mdpi.com/1422-0067/24/18/13922  https://www.mdpi.com/1422-0067/24/18/14197 ) or without italic font (https://www.mdpi.com/1422-0067/24/16/12822  https://www.mdpi.com/1422-0067/24/18/13680) for Firmicutes and Bacteroidetes in particular.

Reviewer 3 Report

Comments and Suggestions for Authors

Review articles, when well done, are interesting, since they group the existing knowledge on a specific subject, and contribute to facilitate the work of other researchers. However, when the review is not complete, or is not done from the right perspective, the attractiveness of this type of article decreases significantly. The work presented here is not particularly attractive, as it lacks depth. Multiple aspects are touched upon, but none of them is adequately developed, and most of them remain at quite superficial levels. The authors opt, in most cases, for an excessively descriptive narrative, in which the studies of other authors form the core of the review, instead of being the support for the facts presented. To be published, this work would have to be written differently, going from facts to examples, and not the other way around.

As a note, I would recommend that the authors take care with the citations of the different taxonomic levels. Apart from the fact that the nomenclature of the bacterial phyla is not up to date, the use of italics is practically random.

Author Response

Dear Editor and Reviewers!

The original manuscript was revised to address the topic of potato microbiome research, emphasizing the relationship with environmental factors and approaches to microbiome modulation. We found no previous reviews dedicated to potato microbiome modulation or related topics. Therefore, the aim of our review is to summarize recent studies on this topic across spectrum of factors (drought, pathogens, soil, etc.)

Our current project focuses on the modulation of the potato microbiome by the biofertilizer Rhodococcus qingshengii VKM Ac-2784D (https://rscf.ru/project/23-26-10049/). The introduction has been revised to more clearly connect the body sections and to show the interrelation of their results with the main objective of review. A figure has been added to the manuscript to demonstrate the relationship between the potato microbiome impact factors discussed in the sections and the practices of microbiome engineering. The conclusion was supplemented by findings and perspectives concerning the development of potato microbiome research. All sections of the manuscript were expanded with proper references according to reviewers’ comments.

Reviewer 3

Q: - 1: The work presented here is not particularly attractive, as it lacks depth. Multiple aspects are touched upon, but none of them is adequately developed, and most of them remain at quite superficial levels. The authors opt, in most cases, for an excessively descriptive narrative, in which the studies of other authors form the core of the review, instead of being the support for the facts presented. To be published, this work would have to be written differently, going from facts to examples, and not the other way around.

A: The aim of our review is to summarize recent studies on this topic across spectrum of factors (drought, pathogens, soil, etc.) and potato plant microbiome modulation. We agree, that there are different ways to organize the review, but depth requires narrowing the topic. Sections 4-7 begin with an introductory part to introduce the reader to the context.

We have revised the introduction and conclusion to emphasize the main purpose of the study and to show the relationship between the theme and results of the paragraphs and the key aspects of potato microbiome studies and engineering. A figure has been added to make this relationship clearer. The paragraphs have also been modified to add a more logical and coherent flow of the text. The conclusion section was extended with perspectives.

Q: - 2: As a note, I would recommend that the authors take care with the citations of the different taxonomic levels. Apart from the fact that the nomenclature of the bacterial phyla is not up to date, the use of italics is practically random.

A: The citations of different taxonomic levels and the use of italics have been corrected according to the International Code of Nomenclature of Prokaryotes. Prokaryotic Code (2022 Revision) (Oren A, Arahal DR, Göker M, Moore ERB, Rossello-Mora R, et al, 2023; https://www.microbiologyresearch.org/content/journal/ijsem/10.1099/ijsem.0.000778?crawler=true)

Since this paper is a review and not a taxonomic study, we believe the most correct approach is to use the nomenclature of the bacterial phyla that was specified in the original publications. Unfortunately, revision of the nomenclature is quite frequent, causing various disputes and disagreements, and it is not possible to rewrite previously published articles each time a revision is made. Even if we now update the nomenclature in the review according to the current state, it will also cease to be relevant in the foreseeable future after the new revisions. Thus, it seems that the most reasonable approach would be to retain the nomenclature used by the authors in the publications to which we refer.

Moreover, we see the papers in IJMS with (https://www.mdpi.com/1422-0067/24/18/13922  https://www.mdpi.com/1422-0067/24/18/14197 ) or without italic font (https://www.mdpi.com/1422-0067/24/16/12822  https://www.mdpi.com/1422-0067/24/18/13680) for Firmicutes and Bacteroidetes in particular.

Round 2

Reviewer 1 Report

Comments and Suggestions for Authors I believe that after taking into account both substantive and linguistic corrections, the article is suitable for publication.           ​                  

Author Response

Dear Editor and Reviewers!

We thank the reviewers for useful comments that help to improve the manuscript.

Our manuscript was revised according to the reviewers’ comments. The target audience of our review is broader than potato microbiome specialists, that therefore each section contains the introductory part followed by the summary of key studies.

As mentioned before, our current project focuses on the modulation of the potato microbiome by the biofertilizer Rhodococcus qingshengii VKM Ac-2784D (https://rscf.ru/project/23-26-10049/). The introduction has been extended to include important very recent study of potato microbiome across several US field sites (https://doi.org/10.1094/PBIOMES-07-23-0060-R) and AMF inoculation case. A figure after the Intro demonstrates existing approaches to study the potato host plant and its microbiome. The conclusion was extended by major findings on the potato microbiome, summarized in Table 2 and perspectives on the development of potato microbiome research.

Reviewer 1

Q: - 1: I believe that after taking into account both substantive and linguistic corrections, the article is suitable for publication.

A: The English language of the manuscript was checked by an expert.

Reviewer 3 Report

Comments and Suggestions for Authors

The authors have introduced minimal changes, which include two to three additional paragraphs, and certain modifications in the wording. These changes incorporate more information, but do not alter the quality of the article. It continues to be excessively descriptive, given that, for the most part, the authors report what they have found in the literature, so that the review becomes a succession of extraneous results, instead of an update of knowledge on the subject under study.

On the other hand, and in relation to nomenclature, italics are used only from the taxonomic level of genus onwards. Anything else is an error, whether or not it appears in other articles. To be aware of the error and continue to maintain it does not seem very scientific. And this same argument can be applied to the updating of bacterial phyla. Obviously, previously published articles cannot be changed, but a text that has not yet been published, and that is under revision, should adopt the new nomenclature.

Comments on the Quality of English Language

In the new contents introduced there are some drafting errors:

"There is a growing interest in crop production is microbiome modulation by biological agents".

- "This is consistent with previous studies showed that rhizosphere populations decreased as the plants mature".

-"The well-known soil-borne pathogen that is caused by Verticillium dahlia...".

-"Authors performed the large-scale experiment...".

Author Response

Dear Editor and Reviewers!

We thank the reviewers for useful comments that help to improve the manuscript.

Our manuscript was revised according to the reviewers’ comments. The target audience of our review is broader than potato microbiome specialists, that therefore each section contains the introductory part followed by the summary of key studies.

As mentioned before, our current project focuses on the modulation of the potato microbiome by the biofertilizer Rhodococcus qingshengii VKM Ac-2784D (https://rscf.ru/project/23-26-10049/). The introduction has been extended to include important very recent study of potato microbiome across several US field sites (https://doi.org/10.1094/PBIOMES-07-23-0060-R) and AMF inoculation case. A figure after the Intro demonstrates existing approaches to study the potato host plant and its microbiome. The conclusion was extended by major findings on the potato microbiome, summarized in Table 2 and perspectives on the development of potato microbiome research.

Reviewer 3

Q: - 1: The authors have introduced minimal changes, which include two to three additional paragraphs, and certain modifications in the wording. These changes incorporate more information, but do not alter the quality of the article. It continues to be excessively descriptive, given that, for the most part, the authors report what they have found in the literature, so that the review becomes a succession of extraneous results, instead of an update of knowledge on the subject under study.

A: The aim of this manuscript is to offer a comprehensive overview of the main aspects pertaining to the potato microbiome and demonstrate the potential benefits that microbiome modulation can bring to modern agriculture. We have revised the logic of each section. Each fact described is accompanied by an appropriate reference and summary from the original study (where applicable). There are many cases, where we checked the consistency between studies, holding critical view to the results. However, many plant-microbe interaction mechanisms remain unclear, which we highlight in the following sections and conclusion. To gain new knowledge, we have to perform the meta-analysis (using raw data from several experiments), which is difficult because of methodological differences between studies.

Q: - 2: On the other hand, and in relation to nomenclature, italics are used only from the taxonomic level of genus onwards. Anything else is an error, whether or not it appears in other articles. To be aware of the error and continue to maintain it does not seem very scientific. And this same argument can be applied to the updating of bacterial phyla. Obviously, previously published articles cannot be changed, but a text that has not yet been published, and that is under revision, should adopt the new nomenclature.

A: We revised all taxonomic group names via database (https://lpsn.dsmz.de/) according to the International Code of Nomenclature of Prokaryotes (https://doi.org/10.1099/ijsem.0.005585) for prokaryotes, and via GBIF for eukaryotes. Synonyms and obsolete names have been changed to correct ones. Italic type has been used to highlight the taxon of family, genus and species rank.

Q: - 3: In the new contents introduced there are some drafting errors:

"There is a growing interest in crop production is microbiome modulation by biological agents".

- "This is consistent with previous studies showed that rhizosphere populations decreased as the plants mature".

-"The well-known soil-borne pathogen that is caused by Verticillium dahlia...".

-"Authors performed the large-scale experiment...".

A: All these sentences were corrected. The English language of the manuscript was checked by an expert.

Round 3

Reviewer 3 Report

Comments and Suggestions for Authors

This third version still does not improve this review work. More text is incorporated, which only contributes to making the article more monotonous. The authors continue to offer an excessively descriptive work. Basically, it is an enumeration of results.

Comments:

- The new name for the phylum Actinobacteria is Actinomycetota, not Actinomycetes. And for Saccharibacteria, Candidatus Saccharimonadota.

- L. 165: Although it is obvious what is being referred to, it is necessary to specify which gap is being filled.

Comments on the Quality of English Language

In general, the use of the articles is quite poor.

Specific comments:

- L. 53-54: The verb is missing.

- L. 70; 310: "Effect" does not seem the most appropriate verb.

- L. 128: The two verbs (mitigating and supressing) should be in the gerund.

- L. 459: "including through".

Author Response

Dear Editor and Reviewer 3

We thank the reviewer for thorough work and useful comments which helped to improve the manuscript.

Reviewer 3

Q: - 1: This third version still does not improve this review work. More text is incorporated, which only contributes to making the article more monotonous. The authors continue to offer an excessively descriptive work. Basically, it is an enumeration of results.

A: The aim of this manuscript is to provide a comprehensive overview of the key aspects pertaining of the potato microbiome and to introduce the broad audience into the topic.

Q: - 2: The new name for the phylum Actinobacteria is Actinomycetota, not Actinomycetes. And for Saccharibacteria, Candidatus Saccharimonadota.

A: All occurrences of phylum Actinomycetes were changed to Actinomycetota. We agree, that name of Saccharibacteria was changed recently to Candidatus Saccharimonadota (changed in the manuscript) (https://doi.org/10.1099/ijsem.0.005821), but there are still no studies using this name (according to Google Scholar and Pubmed)

Q: - 3: Specific comments:

- L. 53-54: The verb is missing.

- L. 70; 310: "Effect" does not seem the most appropriate verb.

- L. 128: The two verbs (mitigating and supressing) should be in the gerund.

- L. 459: "including through".

A: All these sentences were revised for clarity.

Q: - 4: In general, the use of the articles is quite poor.

A: The use of the articles has been verified by the English expert service.
